# Dynamics of growth, physiology, radiation interception, production, and quality of autumn black gram (*Vigna mungo* (L.) Hepper) as influenced by nutrient scheduling

Purabi Banerjee[1], Visha Kumari Venugopalan[2]*, Rajib Nath[1], Ahmed Gaber[3], Akbar Hossain[4]*

1 Department of Agronomy, Bidhan Chandra Krishi Viswavidyalaya, Mohanpur, Nadia, West Bengal, India, 2 Division of Crop Sciences, ICAR-Central Research Institute for Dryland Agriculture, Hyderabad, Telangana, India, 3 Department of Biology, College of Science, Taif University, Taif, Saudi Arabia, 4 Division of Soil Science, Bangladesh Wheat and Maize Research Institute, Dinajpur, Bangladesh

* visha.venugopal@gmail.com (VKV); akbarhossainwrc@gmail.com (AH)

**Data Availability Statement:** All relevant data are within the paper.

## Abstract

To analyse the effect of nutrient management on the growth, physiology, energy utilization, production and quality of black gram, a field trial on black gram was conducted at eastern Indian Gangetic alluvium during the autumn of 2020 and 2021. Treatments were two soil applications of cobalt (Co) and foliar spray of potassium (K) and boron (B) in five combinations. All treatments were arranged in a split-plot design and repeated three times. Two soil applications of cobalt (Co) were assigned in the main plots and foliar spray of potassium (K) and boron (B) in five combinations were assigned in sub-plots. Applications of Co in soil and foliar K+B facilitated significantly higher ($p \leq 0.05$) values for aerial dry matter (ADM), leaf area index (LAI), nodules per plant, total chlorophyll, net photosynthetic rate and nitrate reductase content in both 2020 and 2021, with a greater realization of photosynthetically active radiation interception, and use efficiency (IPAR and PARUE respectively), seed yield, seed nutrients and protein contents. Differences in LAI exhibited positive and linear correlation with IPAR explaining more than 60% variations in different growth stages. The innovative combination of soil Co (beneficial nutrient) application at 4 kg ha$^{-1}$ combined with foliar 1.25% K (macronutrient) + 0.2% B (micronutrient) spray is a potential agronomic management schedule for the farmers to sustain optimum production of autumn black gram through substantial upgradation of growth, physiology, energy utilization, production and quality in Indian subtropics.

## Introduction

The interception of solar radiation holds an immense significance on developmental aspects of crop plants including food legumes [1, 2]. In fact, several shreds of evidence in the case of legumes support acceleration in biomass production in parallel with the increasing rate of

**Funding:** The study was financially supported by Bidhan Chandra Krishi Viswavidyalaya, Mohanpur-741252 by grants awarded to PB, Nadia, West Bengal, India; Bangladesh Wheat and Maize Research Institute, Dinajpur 5200 by grants awarded to AH, Bangladesh and ICAR-Central Research Institute for Dryland Agriculture, Hyderabad 500059, Telangana, India by grants awarded to VK. The research was also funded by Taif University, Saudi Arabia, Project No. (TU-DSPP-2024-07) by grants awarded to AG. The funders had no role in study design, data collection and analysis, decision to publish, or preparation of the manuscript.

**Competing interests:** The authors have declared that no competing interests exist.

interception of the photosynthetically active portion of incoming solar radiation [3]. In other words, proper improvement in crop growth supports appropriate synchrony between its vegetative and reproductive growth through the potential capture of photosynthetically active radiation (PAR) necessary for the eventual realization of optimum yield [4]. Further research efforts specified that the intercepted PAR (IPAR) by a crop canopy is primarily governed by the leaf area index (LAI) in addition to its canopy architecture [5]. Basically, leaf area is a principal determinant of IPAR as well as its utilization in the course of dry matter accumulation and photosynthetic activity [6]. Extensive expansion of leaf area ensures an overall modulation of the developmental rate of the concerned crop and facilitates magnificent interception of PAR, eventually contributing to spectacular economic harvests [7]. In this context, chlorophyll content is one of the elementary attributing characteristics of the leaf physiology of a plant in connection with its photosynthetic capacity. Enhanced biosynthesis of leaf chlorophyll invariably contributes to the capture of a greater amount of incoming solar radiation and a consequent higher rate of net photosynthesis [8]. Evidently, radiation use efficiency is stimulated by the advancement in the interception of PAR [9]. However, improved efficiency of radiation use vividly indicates a better net photosynthetic rate, which in turn fundamentally promotes higher efficiency of production and nutrient use [10].

Black gram [*Vigna mungo* (L.) Hepper] is an important short-duration warm-season food legume cultivated in Indian subtropics [11]. This crop possesses a remarkable capacity for symbiotic fixation of atmospheric nitrogen along with significant restoration properties of soil fertility. Seeds of black gram are supposed to be an excellent reserve in terms of carbohydrate, fat, protein, fibre as well as several minerals and vitamins. The functional aspects of cobalt (Co), potassium (K) and boron (B) have been emphasized in triggering the growth and development of legume crops as a whole with special reference to black gram [2]. Among these nutrient elements, Co is associated with the production of leghaemoglobin protein prerequisite for rhizobial functioning and subsequent nitrogen fixation in legumes facilitating a profound impact on enzymatic activities [12]. Besides, K is a well-known catalytic agent for the activation of several enzyme systems in addition to modulating assimilate transportation and osmotic adjustments inside the plant systems [13]. Likewise, B is involved with several mechanisms of pulse crops including carbohydrate transportation, photosynthetic behaviour, pollen germination together with reproductive growth [14]. In connection with the nitrogen metabolism process for the legumes, nitrate reductase (NR) enzyme activity is much essential for the legume crops with special reference to black gram [15] when the biological nitrogen fixation in the root nodules through the rhizobial functioning gets deactivated, which is most prominent at the time of active development of reproductive sinks [16]. Interestingly, the efficiency rate of a crop concerning the conversion of intercepted solar radiation into plant biomass is likely to be determined by its physiological properties along with some external factors like nutrient availability [17]. In a similar line, a synergistic relationship has been observed between the availability of nutrients to a crop with its radiation use efficiency [18].

Research studies have identified specific deficiencies and growth limitations for Co, K and B in pulse crops, specifically in case of black gram. Inadequate cobalt levels have been reported to reduce nodulation and nitrogen fixation [19], as well as obstructed root development, overall growth and yield [20] in black gram. Studies by Hussain et al. [21] underscored the pivotal role of K in black gram, highlighting its critical role in maintaining physiological processes and overall yield. Channabasava et al. [22] and Devi et al. [23] emphasized the critical impact of B on reproductive processes, yield components and overall crop productivity in black gram. These observation from previous research works have prompted the current investigation to explore the combined effect of these three elements on black gram with a single study.

Taking these background into consideration, a trial was formulated to execute during two subsequent autumn seasons of 2020 and 2021 in view of a hypothesis that soil application of beneficial element (Co) and foliar nutrition of macro (K) with micro (B) nutrients would be an innovative blend of agronomic interventions in upscaling the growth and production autumn sown black gram while sustaining the physiological development as well as the quality of yield for the marginal farmers targeting the intensification of radiation use efficiency. A notable research gap on the influence of this exclusive nutrient schedule on the autumn-grown black gram in Indian subtropics managing the dynamics of radiation utilization necessitated the framework of this trial. The key objectives of this study were to find out the positive role of those nutrient elements on growth, physiology, production and seed quality; to evaluate the impact of nutrient management strategies on interception and use efficiency of solar radiation and to analyze the interconnection of radiation utilization property with the pattern of growth of black gram during the autumn season.

## Materials and methods

### Details regarding the experimental site

The two-year field experiment on black gram (variety: Pant Urd 31) was carried out during the autumn seasons of 2020 and 2021 at the 'A-B' block, District Seed Farm (22°93' N latitude and 88°53' E longitude) of Bidhan Chandra Krishi Viswavidyalaya in Kalyani, Nadia of West Bengal, India. The selected experimental location is situated at 9.75 m altitude above the mean sea level on a uniform topography. The soil of the selected site was medium fertile, well-drained, and nearly neutral (pH: 7.41) Gangetic alluvium of order: *Inceptisol*, technically classified under the sandy loam category containing 65.36% sand, 18.25% silt and 16.32% clay as estimated by Hydrometer method. Just before conducting the experiment, the initial soil of the experimental site was consisting of bulk density (Core method), organic carbon (Wet oxidation method), available nitrogen (modified Kjeldahl method), phosphorus ($0.5\ M$ $NaHCO_3$ extract method), potassium (neutral normal $NH_4OAc$ extract method), cobalt (EDTA extractable method) and boron (Azomethine H method) in the range of $1.25\ g\ cm^{-3}$, 0.51%, 261.63 kg ha$^{-1}$, 39.23 kg ha$^{-1}$ and 195.72 kg ha$^{-1}$, 0.56 ppm and 0.48 ppm respectively.

### Meteorological data

The daily readings regarding meteorological parameters i.e., maximum and minimum temperatures, rainfall and bright sunshine hours at the study site for the entire experimental period from August 2020 to November 2020 and August 2021 to October 2021 were collected from All India Coordinated Research Project on Agrometeorology (AICRPAM), BCKV, Nadia, West Bengal. The average values encompassing maximum and minimum temperatures, rainfall and bright sunshine hours during different growth stages of autumn sown black gram covering both the years of experimentation (2020 and 2021) have been exhibited in Table 1 and Fig 1.

### Experimental design and treatment details

In accordance with the objectives of the experiment, the field trial was laid out in a split-plot design and replicated thrice. Two different levels of Co application i.e., no cobalt application and application of Co [Co (NO$_3$)$_2$ 6 H$_2$O (Co: 21%) @ 4 kg ha$^{-1}$] were allotted to the main plots, while different foliar application levels of 1.25% K and 0.2% B at flower initiation stage including no spray, tap water, K @ 1.25% [as Mureate of Potash (K$_2$O: 60%)], B @ 0.2% [as Borax (B: 11.5%)] and combined K @ 1.25% + B @ 0.2% were assigned in subplots. The entire

**Table 1. Stage-wise mean temperature (maximum and minimum), rainfall and bright sunshine hours during black gram growing season (autumn) in 2020 and 2021.**

| Parameter | Maximum temperature (°C) | | Minimum temperature (°C) | | Rainfall (mm) | | Bright sunshine hours (hour) | |
|---|---|---|---|---|---|---|---|---|
| Stages | 2020 | 2021 | 2020 | 2021 | 2020 | 2021 | 2020 | 2021 |
| Sowing | 31.94 | 33.19 | 26.54 | 26.54 | 7.44 | 7.00 | 3.20 | 3.85 |
| Flowering | 32.55 | 31.57 | 26.25 | 25.67 | 7.15 | 9.90 | 4.17 | 4.47 |
| Pod filling | 33.17 | 32.70 | 25.92 | 25.63 | 12.37 | 13.37 | 5.79 | 5.43 |
| Maturity | 32.07 | 30.15 | 23.72 | 23.60 | 0.69 | 7.90 | 7.00 | 4.55 |

treatment allotment process followed a randomised manner to reduce errors. Each plot was sized into a 4 m × 3 m area, which led to a gross experimental area of 500 m$^2$ with 60 m$^2$ of main plots.

## Crop management practices

Black gram seeds of variety Pant Urd 31 were sown in individual treatments maintaining 30 cm of row spacings. The variety Pant Urd 31 was selected in this respect considering its higher production potential along with excellent adaptability under the Indian climatic scenario [24]. Recommended basal dosages of inorganic fertilizers for black gram crop including N, P$_2$O$_5$ and K$_2$O were applied respectively at 20, 40 and 40 kg ha$^{-1}$ during final land preparation but before seed sowing. Emerged seedlings were thinned after 10–15 days to maintain plant-to-

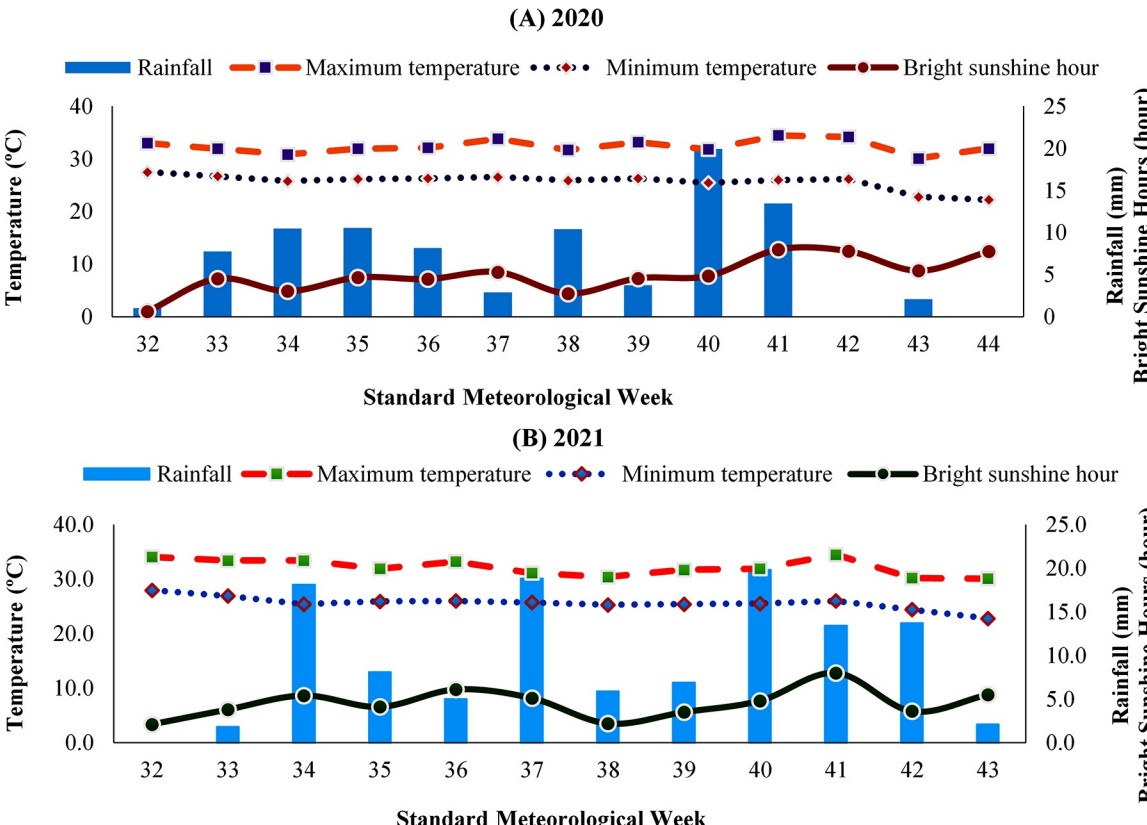

**Fig 1.** Mean weekly distribution of meteorological features of black gram growing period during autumn seasons of (A) 2020 and (B) 2021.

plant distance of 10 cm to ensure optimum plant population. The crop stand was kept weed-free up to its critical period of the first 25–30 days from sowing. Foliar spray operations were performed within the early morning hours at the flower initiation stage following the nutrient allotment schedules (treatment-wise). Nutrients were sprayed with knapsack sprayers by a single labourer each time simply walking through the individual plots. The crop was grown solely under rainfed conditions in the autumn season of both years, i.e., no external application of irrigation was provided. Spraying of Chlorantraniliprole (insecticide) 18.5 SC @ 0.2 ml lit$^{-1}$ of water and a mixture of Mancozeb + Carbendazim (fungicide) @ 2.5 g lit$^{-1}$ of water at 25 days and 45 days after sowing were employed to protect the crop from insect and disease damages.

## Estimation of crop growth, physiology, intercepted solar radiation, yield and quality parameters

Five randomly tagged plants were used to observe various growth parameters from each plot excluding the border rows. Aerial dry matter (ADM), leaf area index (LAI) and number of root nodules per plant were estimated by collecting samples at vegetative (15 DAS), flowering (30 DAS) pod filling (45 DAS) and at maturity (65 DAS) stages.

For the estimation of ADM, samples from aerial portions of the plants were dried in a hot air oven at 80˚C temperature for 24–48 hours upto constant weights and calculated using the following Eq 1:

$$\text{ADM (g m}^{-2}) = \frac{\text{Dry weight of aerial part per plant (g)} \times \text{Number of plants}}{\text{Ground area (m}^2)} \qquad (1)$$

Regarding LAI determination, green leaves were separated from the stem to measure the mean leaf area of a single plant and the LAI values were derived using the following Eq 2 [25]:

$$\text{LAI} = \frac{\text{Leaf area per plant (m}^2) \times \text{Number of plants}}{\text{Ground area (m}^2)} \qquad (2)$$

The total chlorophyll content in mg g$^{-1}$ of fresh weight of black gram leaf was measured at 50% flowering stage by following the formula (Eq 3) given by Arnon [26] after taking the absorbance readings at different wavelengths including 645 and 663 nm against a blank sample with 80% acetone in a spectrophotometer (Systronics-105).

$$\text{Total chlorophyll} = (20.2 \times A_{645}) + (8.02 \times A_{663}) \times V/W \times 1000 \qquad (3)$$

Where, V: Extract volume (ml); W: Fresh weight of leaf tissue (g); A: Absorbance

The net photosynthetic rate in µmol m$^{-2}$ sec$^{-1}$ was recorded at 50% flowering stage of black gram preferably on clear sunny days from the last fully developed upper leaves of randomly selected five plants from 11:30 a.m. to 12.30 p.m. at vegetative (15–20 DAS), flowering (30–35 DAS) pod filling (45–50 DAS) and at maturity (65–70 DAS) stages using a portable handheld photosynthesis system (CI-340 Handheld Photosynthesis system, CID Bio-Science, Inc. Camas, WA, USA). Studying the net photosynthetic rates at 50% flowering stage in black gram is crucial due to the energy demand for reproductive growth and resource allocation to flower and pod development. This stage signifies a metabolic shift towards reproductive phases, emphasizing the plant's ability to support these processes. Monitoring photosynthesis provides insights into the plant's physiological status and allows for timely interventions to optimize productivity. Such understanding is vital for improved crop yield outcomes [27].

Nitrate reductase content (µmol g$^{-1}$ fresh leaf weight hour$^{-1}$) in freshly plucked leaf tissues at 50% flowering stage was determined by following the procedure indicated by Singh and Nair [28]. 250 mg of fresh leaf tissue was taken, rinsed in cold distilled water thoroughly and

cut in small pieces with blade in ice bath. The pieces were suspended properly in 5 ml of medium taken in clean test tube. The medium contained potassium phosphate buffer (pH 7.6), 0.1 M $KNO_3$, n-propanol, chloramphenicol and distilled water. The tubes were kept in dark after sealing at 25°C for 25 minutes. Then 2 ml of aliquots was taken from both sample and blank tube separately. 1 ml of 1% sulphanilamide was added in each tube and mix properly. After that, 1 ml of 0.02% N(-1- Napthyl) ethyleneamine dihydrochloride (NED) was added in each case with thorough mixing. After 10 minutes, the contents were diluted by addition of 1 ml distilled water and finally, the pinkish brown was read against blank at 540 nm wavelength in UV-Vis spectrophometer.

$$V = \frac{(T - B)}{W \times t} \qquad (4)$$

Where, V: μmol of $KNO_2$ (μmol $g^{-1}$ fresh weight $hour^{-1}$); T: Concentration of treatment absorbance; B: Concentration of blank absorbance; W: Weight of leaf sample (g); T; Incubation period (hour).

NR activity was investigated at the 50% flowering stage in black gram to assess peak nitrogen demand during reproductive growth, aiding nutrient management. After flower initiation, legumes limit rhizobial activity, prioritizing photo-assimilate allocation to developing sink organs. Symbiotic nitrogen fixation declines as pods develop, hindering atmospheric nitrogen uptake. Black gram, with its indeterminate growth habit, relies on stored nitrate nitrogen conversion to meet internal nitrogen needs for seed protein synthesis, which is crucial for optimizing crop yield [29].

For estimating the intercepted photosynthetically active radiation (IPAR) in MJ $m^{-2}$, a line quantum sensor (APOGEE Logan UT) was placed 25 cm above the crop canopy in parallel across the row to record the incident radiation. The methodology of Banerjee et al. [2] was adopted to measure IPAR. IPAR (MJ $m^{-2}$) was computed using the following Eq 5 as mentioned by Dhaliwal et al. [30]:

$$IPAR = PAR_{(o)} - PAR_{(t)} - PAR_{(r)} \qquad (5)$$

Where, $PAR_{(o)}$ = incident PAR above the canopy; $PAR_{(t)}$ = transmitted PAR through the canopy to the soil surface; $PAR_{(r)}$ = reflected PAR from the uppermost layer of the crop canopy

This value was then converted into per cent (%) using the following Eq 6:

$$\% \; IPAR = \frac{Interceped \; PAR \; (MJ \; m^{-2})}{Incident \; PAR \; at \; the \; top \; of \; the \; crop \; canopy \; (MJ \; m^{-2})} \times 100 \qquad (6)$$

PAR use efficiency (g $MJ^{-1}$) in terms of ADM was calculated with the following Eq 7 as per Confalone et al. [31]:

$$PARUE \; (g \; MJ^{-1}) = \frac{ADM \; (g \; m^{-2})}{IPAR \; (MJ \; m^{-2})} \qquad (7)$$

The seed yield of black gram was recorded after harvesting and threshing of the crop from each plot (4 m × 3 m) covering all three replications and was converted to kg $ha^{-1}$.

The available nitrogen (N), phosphorous (P) and potassium (K) contents in black gram seeds were estimated respectively by the modified Kjeldahl method, 0.5 M $NaHCO_3$ extract method and neutral normal $NH_4OAc$ extract method. The seed protein content was determined by multiplying the N content with a conversion factor of 6.25.

## Statistical analysis

Statistical analysis was exercised for the data following the analysis of variance (ANOVA) technique specified for split splot design [32]. Critical differences were worked out for the sake of comparing the treatment means in terms of significant differences at a significance level of 5%. In this regard, Tukey's post hoc test was carried out to compare the differences between treatment means. The regression analysis was performed with the help of SPSS 7.5 software (SPSS 7.5 copyright, 1997 by SPSS Inc., USA Base 7.5 Application guide).

## Results

### Effect of soil and foliar application of plant nutrients on growth traits

Soil incorporation of Co helped in the accumulation of the significantly higher amount of ADM right from vegetative stage towards maturity (51.0, 94.5, 165.0 and 242.5 g m$^{-2}$ respectively) in comparison with its no application during 2020 (Fig 2A), which was a prominent reflection in the corresponding LAI values (0.48, 0.87, 2.12 and 3.16 respectively) throughout the stages (Fig 3A). Similar trend was followed in the next year with respect to Co application in case of accumulation of ADM (52.2, 97.1, 166.1 and 244.1 g m$^{-2}$ respectively) and LAI (0.50, 0.89, 2.11 and 3.17 respectively) in the subsequent growth stages of autumn sown black gram (Fig 2B and 3B respectively). Regarding the foliar spray factor, combined foliar spray of K+B recorded maximum ADM (254.6 and 257.3 g m$^{-2}$ respectively) and LAI values (3.24 and 3.22 respectively) irrespective of years, which were statistically significant over the rest foliar levels from pod filling onwards. In line with this, foliar B spray performed significantly better in comparison to K spray in case of ADM production as well LAI at pod filling and maturity respectively during 2020 and 2021.

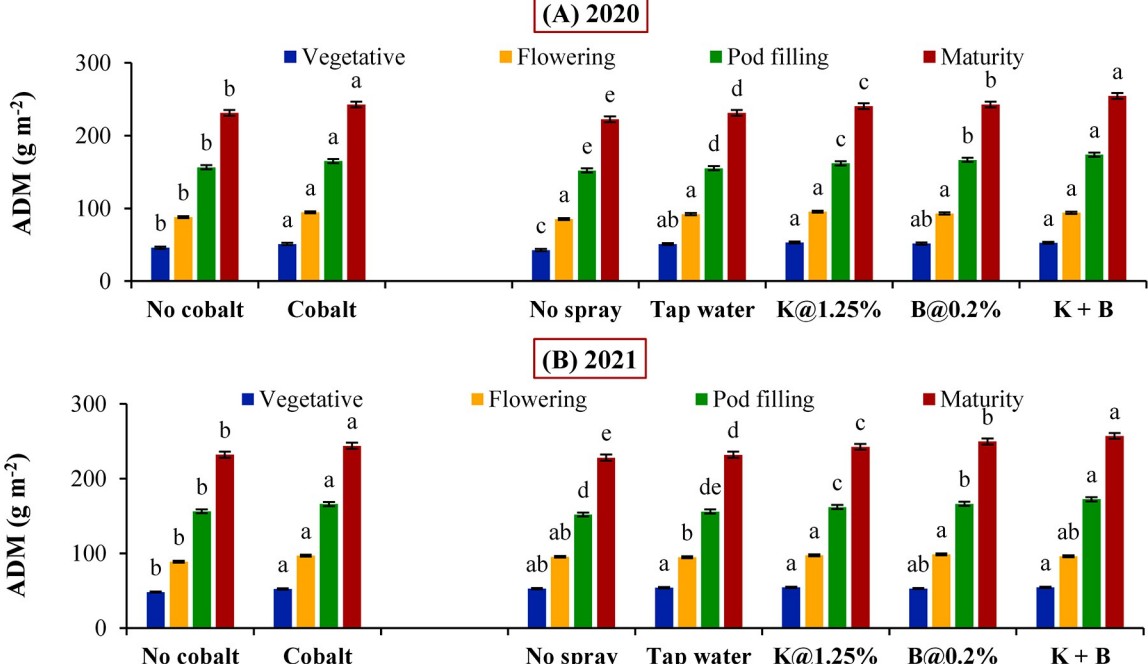

**Fig 2.** Aerial dry matter (ADM) autumn sown black gram in (A) 2020 and (B) 2021 crop seasons as influenced by soil and foliar application of plant nutrients (Error bars designate the standard error of the mean and different alphabets denote significant differences between means).

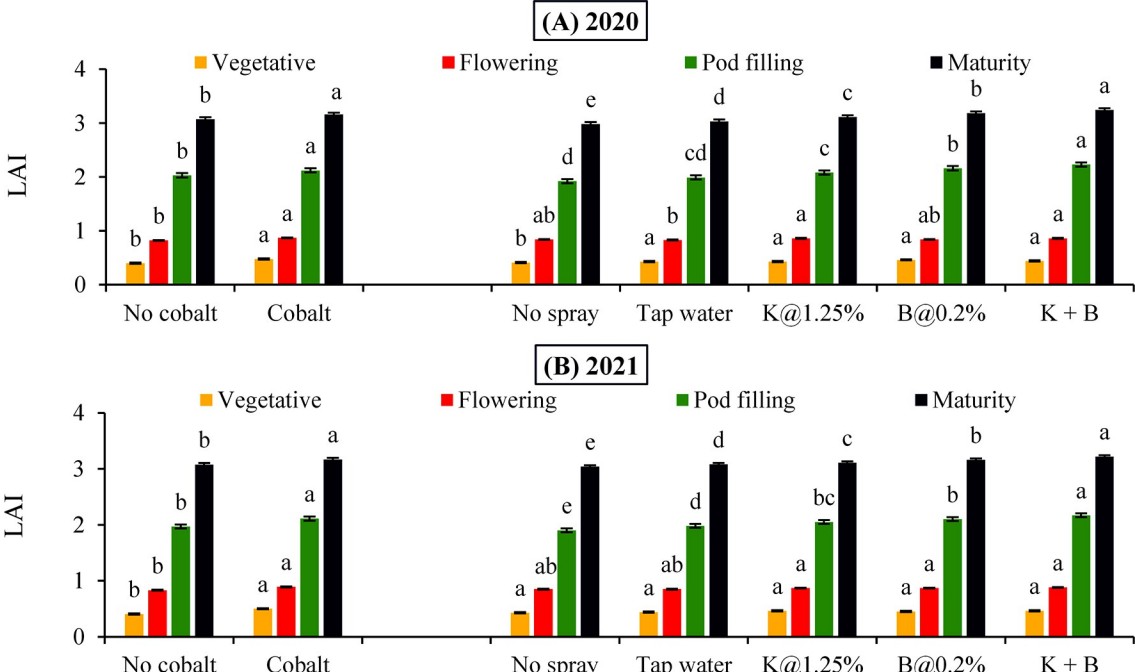

**Fig 3.** Leaf area index (LAI) of black gram in (A) 2020 and (B) 2021 crop seasons as influenced by soil and foliar application of plant nutrients (Error bars exhibit the standard error of the mean and different letters imply differences between means).

## Effect of soil and application of plant nutrients on root nodulation

Stage-wise data pertaining to the number of root nodules per plant of autumn black gram during 2020 and 2021 have been presented respectively in Table 2.

The trend in nodule number is presented in Table 2 for the consecutive experimental years. We observed an increasing trend till flowering in both the years. Soil application of Co

**Table 2. Number of root nodules of autumn sown black gram as influenced by soil and foliar application of plant nutrients in both years.**

| Treatment | 2020 | | | 2021 | | |
|---|---|---|---|---|---|---|
| | Vegetative | Flowering | Pod filling | Vegetative | Flowering | Pod filling |
| Soil application (S) | | | | | | |
| No Cobalt | 27.7±0.04b | 46.5±0.14b | 39.9±0.08b | 30.1±0.05b | 45.8±0.13b | 43.2±0.08b |
| Co @ 4 kg ha⁻¹ | 30.5±0.07a | 53.5±0.11a | 45.3±0.10a | 38.1±0.08a | 51.1±0.16a | 45.1±0.10a |
| LSD (0.05) | 0.19 | 0.46 | 0.26 | 0.15 | 0.52 | 0.38 |
| Foliar Spray (F) | | | | | | |
| No spray | 30.5±0.02ab | 43.3±0.11e | 35.7±0.11e | 31.7±0.03c | 41.8±0.10e | 38.3±0.11e |
| Tap water | 25.5±0.04c | 46.3±0.13d | 39.7±0.12e | 34.0±0.05b | 45.0±0.16d | 40.0±0.12e |
| K @ 1.25% | 31.8±0.01a | 49.8±0.16c | 43.5±0.16c | 35.0±0.04ab | 48.8±0.13c | 44.3±0.16c |
| B @ 0.2% | 26.0±0.07c | 53.0±0.18b | 45.5±0.14b | 34.2±0.04b | 51.5±0.17b | 46.8±0.14b |
| K + B | 31.5±0.03a | 57.7±0.12a | 48.8±0.10a | 35.5±0.06a | 55.0±0.14a | 51.3±0.10a |
| LSD (0.05) | NS | 0.68 | 0.38 | NS | 0.79 | 0.42 |
| Interaction LSD (0.05) | | | | | | |
| S × F | NS | 1.26 | 0.35 | NS | 1.24 | 0.36 |
| F × S | NS | 1.42 | 0.43 | NS | 1.35 | 0.42 |

NS: Non-significant

considerably improved the nodule number at vegetative (30.5 and 38.1), flowering (53.5 and 51.1) and pod filling (45.3 and 45.1) respectively during 2020 and 2021. The active nodulation period produced 57.7 and 55.0 nodules with combined foliar K+B spray during the respective years, which were statistically significant over foliar K (49.8 and 48.8) and B spray (53.0 and 51.5). Additionally, both the years experienced significant interactions at flowering and pod filling stage in this regard.

## Effect of soil and foliar application of plant nutrients on total chlorophyll content and net photosynthetic rate of leaf

Soil application as well as foliar nutrition imposed a remarkable constructive role related to the enhancement of photosynthetic pigment synthesis with respect to total chlorophyll and corresponding net photosynthetic rate in autumn sown black gram leaves during its 50% flowering stage. Incorporation of Co recorded significantly higher total chlorophyll content (1.29 and 1.34 mg $g^{-1}$ of fresh weight of leaf) and net photosynthetic rate (10.46 and 11.38 $\mu$mol $m^{-2}$ $sec^{-1}$) over the treatments without Co respectively across the years (Table 3). Among the foliar treated plots, foliar K and B either as single or in combination achieved greater pigment concentration along with net photosynthetic rate compared to no spray. However, foliar K+B attained significantly higher values of both parameters followed by a single foliar spray of K during the respective years. Accordingly, significant interaction effects were observed among these two factors with respect to both total chlorophyll and net photosynthetic rate.

## Effect of soil and foliar application of plant nutrients on nitrate reductase content

Nitrate reductase content in autumn sown black during 2020 and 2021 are presented in Table 3. The crop attained this enzyme content to the tune of 2.78 and 3.07 $\mu$mol $g^{-1}$ leaf fresh weight $hour^{-1}$, which were statistically significant over no use of Co in the corresponding years. Besides, nitrate reductase content ranged between 2.56–2.84 and 2.84–3.16 $\mu$mol $g^{-1}$ leaf

**Table 3. Physiological characters of autumn sown black gram as influenced by soil and foliar application of plant nutrients in both years.**

| Treatment | 2020 | | | 2021 | | |
|---|---|---|---|---|---|---|
| | Total chlorophyll (mg $g^{-1}$ leaf fresh weight) | Net photosynthetic rate ($\mu$mol $m^{-2}$ $sec^{-1}$) | Nitrate reductase ($\mu$mol $g^{-1}$ leaf fresh weight $hour^{-1}$) | Total chlorophyll (mg $g^{-1}$ leaf fresh weight) | Net photosynthetic rate ($\mu$mol $m^{-2}$ $sec^{-1}$) | Nitrate reductase ($\mu$mol $g^{-1}$ leaf fresh weight $hour^{-1}$) |
| Soil application (S) | | | | | | |
| No Cobalt | 1.24±0.01b | 9.25±0.25b | 2.60±0.02b | 1.29±0.01b | 9.69±0.21b | 2.93±0.02b |
| Co @ 4 kg $ha^{-1}$ | 1.29±0.01a | 10.46±0.28a | 2.78±0.02a | 1.34±0.02a | 11.38±0.36a | 3.07±0.02a |
| LSD $_{(0.05)}$ | 0.03 | 1.02 | 0.09 | 0.03 | 1.27 | 0.06 |
| Foliar Spray (F) | | | | | | |
| No spray | 1.15±0.01e | 6.79±0.59e | 2.56±0.01e | 1.19±0.01e | 0.31±0.42e | 2.84±0.01e |
| Tap water | 1.21±0.01d | 7.34±0.42d | 2.61±0.02d | 1.24±0.03d | 0.34±0.44d | 2.91±0.02d |
| K @ 1.25% | 1.31±0.01b | 8.52±0.61b | 2.76±0.02b | 1.38±0.02b | 0.37±0.63bc | 3.07±0.02b |
| B @ 0.2% | 1.26±0.03c | 8.09±0.53c | 2.68±0.01c | 1.33±0.01c | 0.38±0.51b | 3.00±0.01c |
| K + B | 1.38±0.02a | 9.41±0.57a | 2.84±0.02a | 1.44±0.02a | 0.43±0.52a | 3.16±0.02a |
| LSD $_{(0.05)}$ | 0.04 | 1.74 | 0.07 | 0.03 | 1.58 | 0.05 |
| Interaction LSD $_{(0.05)}$ | | | | | | |
| S × F | 0.02 | 1.35 | 0.02 | 0.02 | 1.48 | 0.02 |
| F × S | 0.03 | 1.57 | 0.02 | 0.03 | 1.83 | 0.03 |

fresh weight hour$^{-1}$ in both years. In each of the years, the treatment with foliar K+B recorded the maximum content of the concerned enzyme, which was statistically significant over the rest of the foliar sprayed treatments. Foliar K was found to be statistically superior to foliar B in this regard respectively during 2020 (2.76 vs. 2.68 µmol g$^{-1}$ leaf fresh weight hour$^{-1}$) and 2021 (3.07 vs. 3.00 µmol g$^{-1}$ leaf fresh weight hour$^{-1}$).

## Effect of soil and foliar application of plant nutrients on interception pattern of photosynthetic active radiation (PAR)

Intercepted photosynthetic active radiation (IPAR) improved over the growth stages of the black gram growth period over the subsequent years (Fig 4). Substantial variation in interception pattern in PAR was recorded with respect to soil and foliar applications of plant nutrients during different phenological stages under observations. The percentage of interception of solar radiation was found to be significantly higher when supplemented with soil application of Co. Regarding the foliar applied treatments, the absence of specific pattern noted during the vegetative as well as flowering stage of the crop justified the implementation of nutrients spray schedule at the very onset of the flowering stage.

Nonetheless, combined foliar K+B at pod filling and maturity stages showed the highest interceptions, which were statistically significant when compared to the remaining treatments in both years. Notably, the efficacy of IPAR for the different treatment combinations was confirmed in relation to LAI at various growth phases, encompassing the vegetative, blooming, pod filling, and maturity stages of black gramme seeded in the autumn. In 2020 and 2021, IPAR and LAI were found to have positive correlations at every stage of growth (Fig 5). Variations in LAI contributed to around 79.89, 75.33, 72.43, and 72.13% changes in 2020 and about 76.68, 72.60, 69.80, and 64.58% in 2021, respectively.

## Effect of soil and foliar application of plant nutrients on PAR use efficiency (PARUE)

The illustration of PAR use efficiency (PARUE) in terms of the accumulation of ADM in different treatments has elucidated marked differences among the treatments right from the vegetative to the maturity stage (Fig 6).

Application of Co significantly influenced the PARUE over its no application during the consecutive experimental years under study. It ranged from 0.68–0.74 g MJ$^{-1}$ at vegetative, 1.15–1.29 g MJ$^{-1}$ at flowering, 1.76–1.81 g MJ$^{-1}$ at pod filling and from 2.17–2.36 g MJ$^{-1}$ at the maturity stage of autumn black gram respectively during 2020 and 2021. Interestingly, significant improvement of PARUE to the tune of 2–4% was observed during the reproductive stage of the crop in the treatments supplemented with foliar K+B nutrition. Although, foliar K and foliar B spray were found to be statistically at par during pod filling stage, foliar B with a PARUE of 2.17 g MJ$^{-1}$ outperformed foliar K spray with the same of 2.16 g MJ$^{-1}$ at maturity during 2020. However, foliar B nutrition recorded significant higher values of PARUE over foliar K spray at both pod filling (1.79 vs. 1.78 g MJ$^{-1}$) and maturity (2.31 vs. 2.29 g MJ$^{-1}$) in the subsequent year of experimentation.

## Effect of plant nutrients on seed yield

The yield of the crop is elucidated in Fig 7. Soil Co application is markedly attributed to the attainment of 12.8 and 6.8% higher economic yield in the respective years in comparison with no use of Co (1275.1 and 1320.4 kg ha$^{-1}$). With the involvement of foliar nutrition, seed yield significantly varied in the range from 1110.6–1586.5 kg ha$^{-1}$ and from 1091.8–1642.7 kg ha$^{-1}$

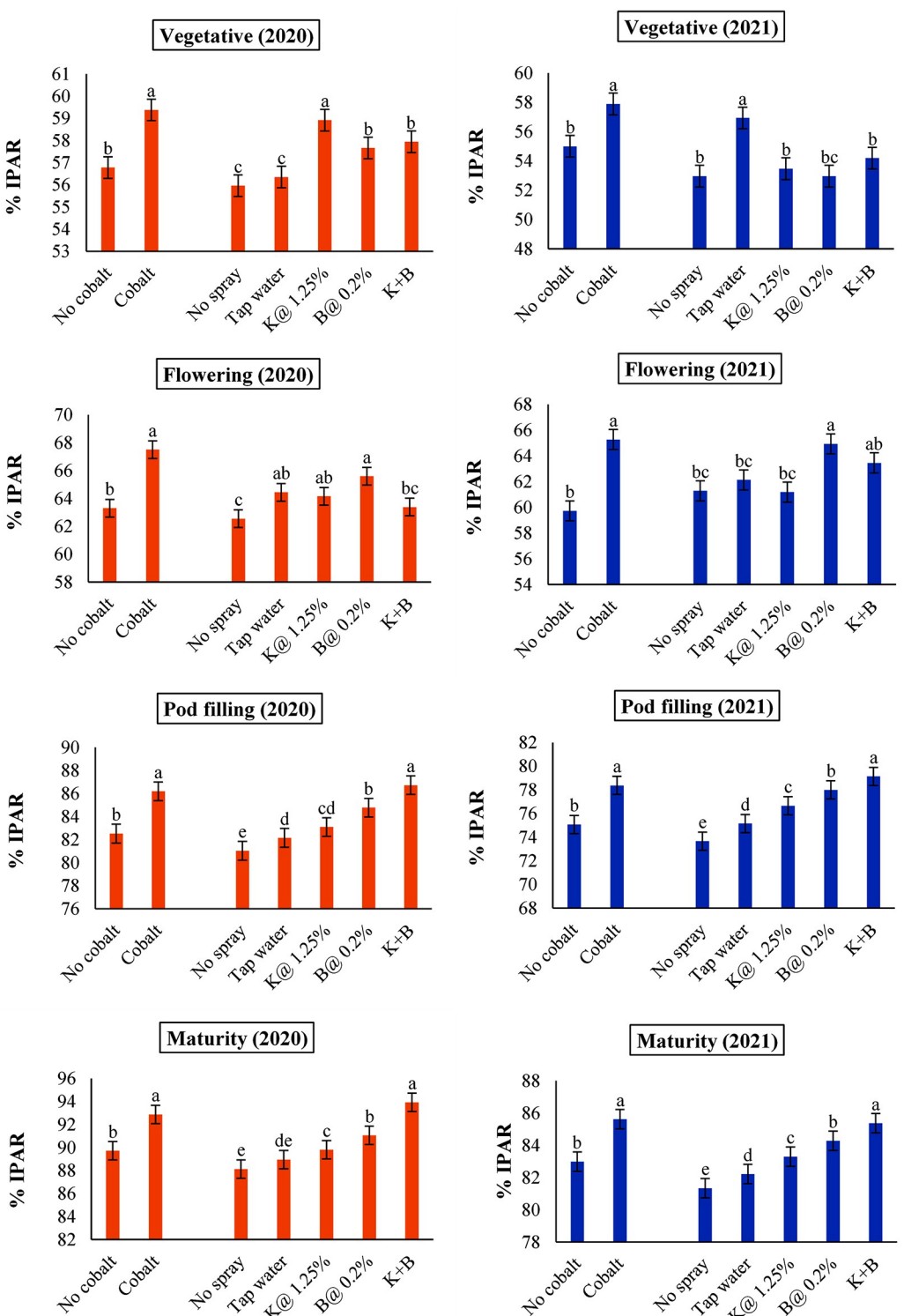

**Fig 4. Intercepted PAR (IPAR) at different stages of autumn sown black gram as influenced by soil and foliar application of plant nutrients in both years (Error bars signify the standard error of the mean.** Different letters designate significant differences between means).

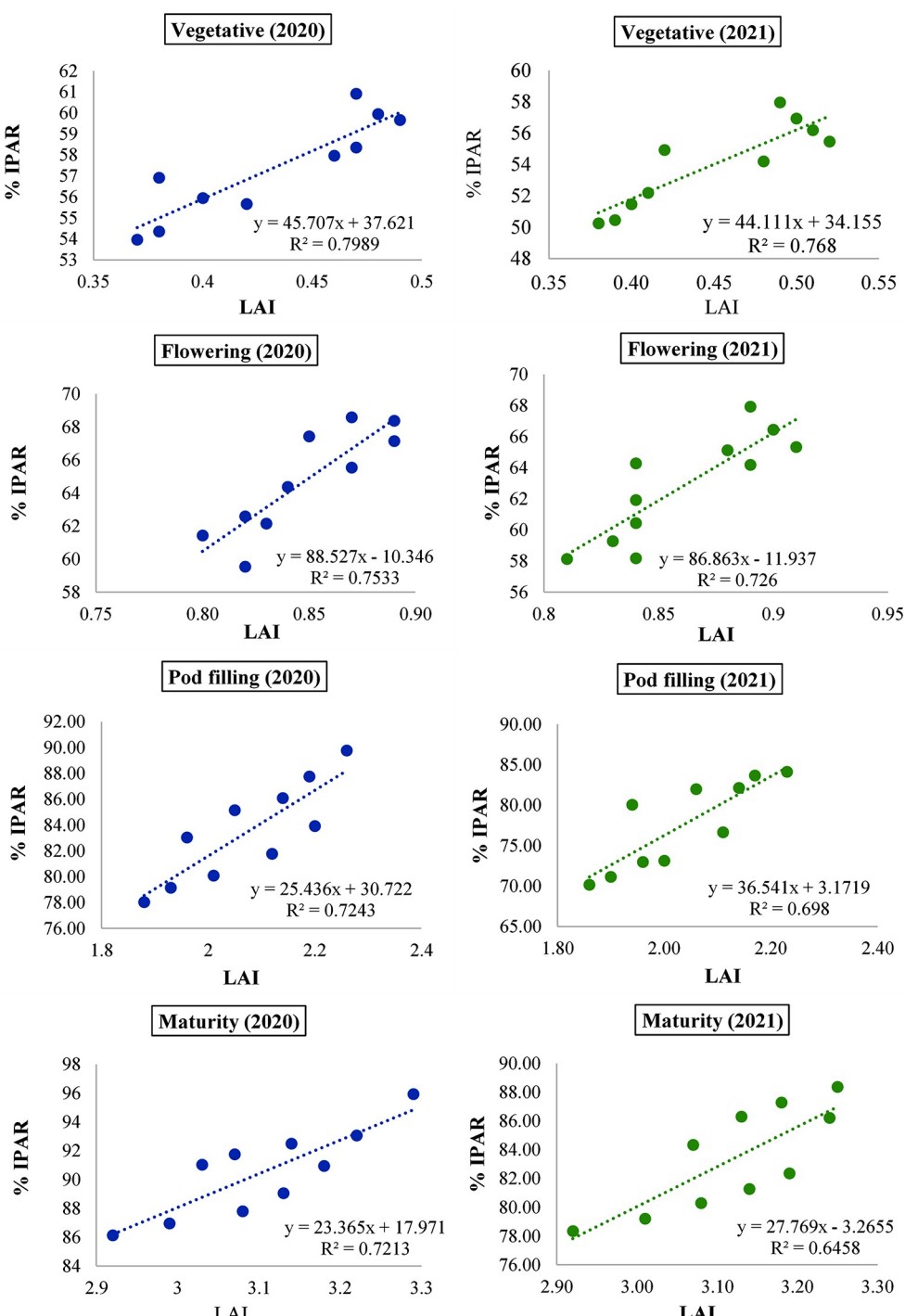

**Fig 5. Relation of LAI with intercepted PAR at different stages of autumn sown black gram as influenced by soil and foliar application of plant nutrients in both years (Different alphabets delineate significant differences between means).**

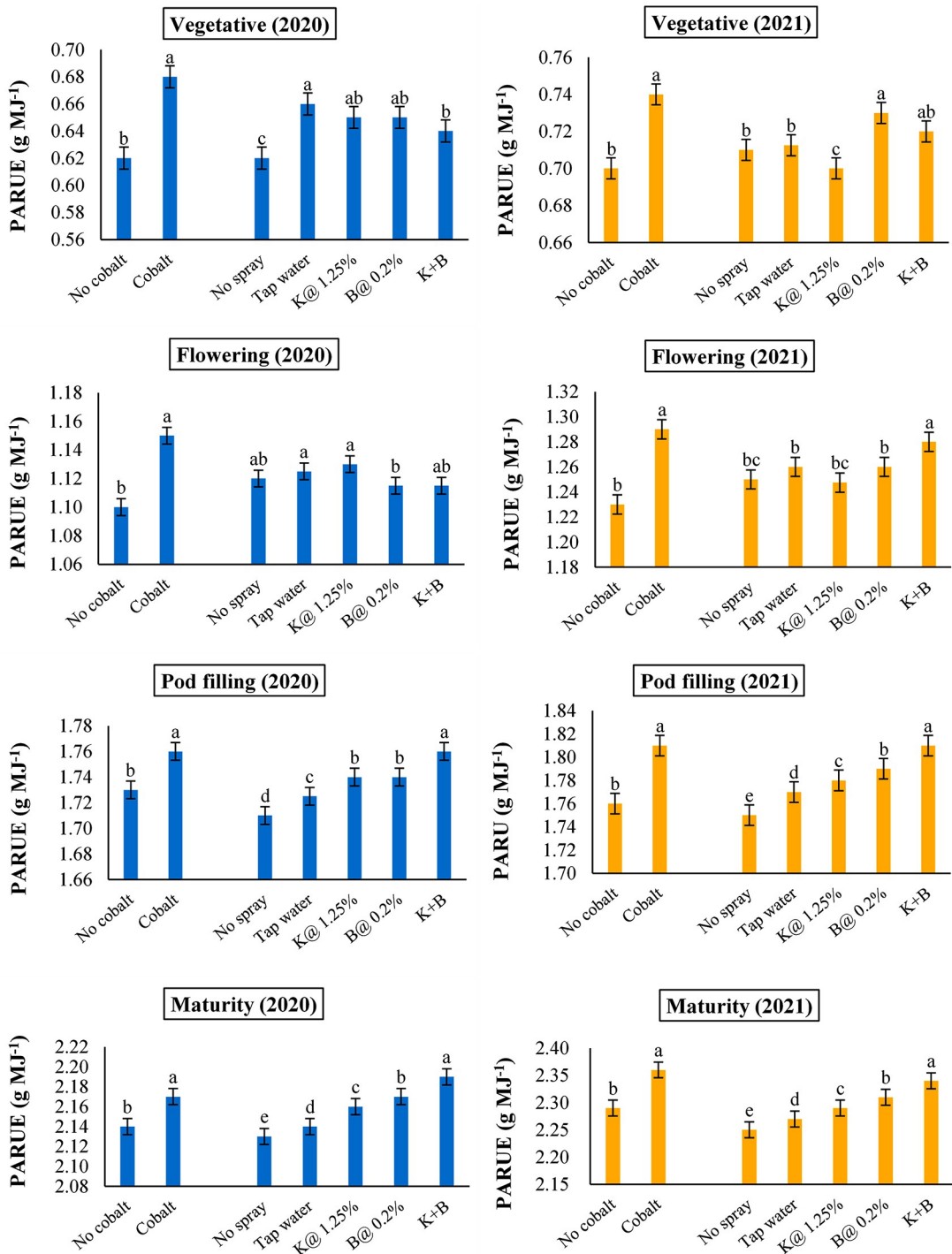

**Fig 6. PAR use efficiency (PARUE) at different growth stages of autumn sown black gram as influenced by soil and foliar application of plant nutrients in both years (Error bars represent the standard error of mean and different alphabets designate significant differences between means).**

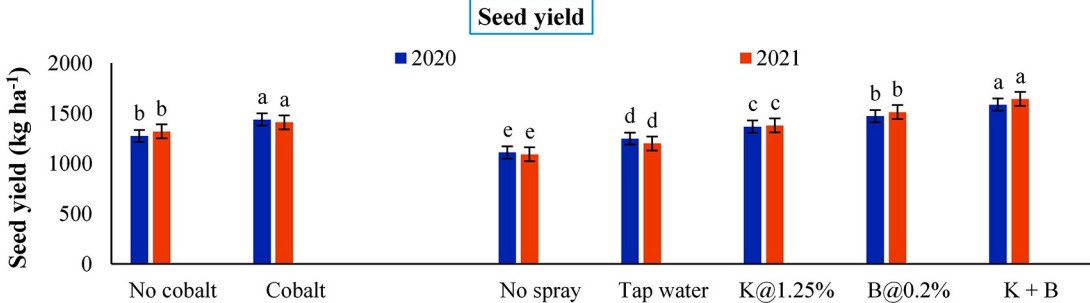

**Fig 7. Seed yield of autumn sown black gram as influenced by soil and foliar application of plant nutrients in both years (Error bars signify the standard error of mean and different alphabets exhibit significant differences between means)**

respectively during 2020 and 2021 obtaining maximum values with K+B spray in each case. Single B spray was registered to be superior to single K spray.

## Effect of soil and foliar application of plant nutrients on major nutrients and protein in seed

There was significant improvement in all the major nutrients like N (4.11%), P (0.29%) and K (1.11%) in the black gram seed during 2020 as well as 1.34, 11.38 and 3.07% for the respective nutrients in the subsequent year (Table 4).

The protein content was also influenced by the nutrient application (Fig 8). In the next year, these values ranged from 3.96, 0.27, 1.08 and 24.87% in terms of the respective parameters, being significantly higher over the treatments without Co application. A combination of K+B had resulted in improved seed quality in all respects during the subsequent years. Foliar B spray recorded significantly higher N (4.15 and 4.02%), P (0.31 and 0.28%) and protein contents (25.92 and 25.15%) over foliar K (3.90 and 3.72%, 0.27 and 0.26% and 24.44 and 23.19% respectively), whereas the reverse was registered only in case of K contents (1.13 and 1.10% vs. 1.10 and 1.07%) in the respective years.

**Table 4. Seed N, P and K contents of autumn sown black gram as influenced by soil and foliar application of plant nutrients in both years.**

| Treatment | 2020 | | | 2021 | | |
|---|---|---|---|---|---|---|
| | N content (%) | P content (%) | K content (%) | N content (%) | P content (%) | K content (%) |
| Soil application (S) | | | | | | |
| No Cobalt | 3.67±0.01b | 0.26±0.01b | 1.09±0.02b | 1.29±0.01b | 9.69±0.21b | 2.93±0.02b |
| Co @ 4 kg ha⁻¹ | 4.11±0.03a | 0.29±0.01a | 1.11±0.01a | 1.34±0.02a | 11.38±0.36a | 3.07±0.02a |
| LSD (0.05) | 0.11 | 0.02 | 0.02 | 0.03 | 1.27 | 0.06 |
| Foliar Spray (F) | | | | | | |
| No spray | 3.48±0.01e | 0.23±0.01e | 1.03±0.01e | 1.19±0.01e | 0.31±0.42e | 2.84±0.01e |
| Tap water | 3.62±0.02d | 0.25±0.02d | 1.06±0.02d | 1.24±0.03d | 0.34±0.44d | 2.91±0.02d |
| K @ 1.25% | 3.90±0.01c | 0.27±0.01c | 1.13±0.01b | 1.38±0.02b | 0.37±0.63bc | 3.07±0.02b |
| B @ 0.2% | 4.15±0.02b | 0.31±0.03b | 1.10±0.02c | 1.33±0.01c | 0.38±0.51b | 3.00±0.01c |
| K + B | 4.31±0.02a | 0.34±0.03a | 1.17±0.02a | 1.44±0.02a | 0.43±0.52a | 3.16±0.02a |
| LSD (0.05) | 0.09 | 0.02 | 0.02 | 0.03 | 1.58 | 0.05 |
| Interaction LSD (0.05) | | | | | | |
| S × F | 0.10 | NS | 0.02 | 0.02 | 1.48 | 0.02 |
| F × S | 0.07 | NS | 0.03 | 0.03 | 1.83 | 0.03 |

NS: Non-significant

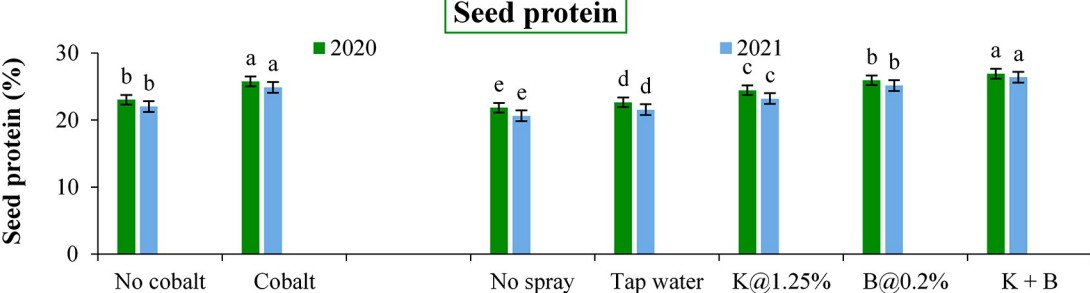

**Fig 8. The seed protein content of autumn sown black gram as influenced by soil and foliar application of plant nutrients (Error bars indicate the standard error of mean and different alphabets delineate significant differences between means).**

## Discussion

Crops typically have restrictions on their vegetative growth after their reproductive development begins. On the other hand, it has been found that legume crops have an indeterminate development pattern that allows for simultaneous vegetative and reproductive growth. Here, the reproductive components acquire the majority of the dry matter. Nevertheless, throughout the reproductive phase, a few fresh leaf flushes will sporadically appear in parallel with the ageing of older leaves. Simultaneously, it emerged that the initial Co-treatment was linked to both limited leaf senescence and increased leaf area expansion [33]. This appeared to have persisted even after the commencement of reproductive growth in a progressive increase in dry biomass accumulation and LAI. Furthermore, foliar spraying at the flower initiation stage proved to be an excellent strategy for allowing black gram to thrive with increased leaf area throughout the reproductive phase. ADM was significantly higher with tap water spray compared to no spray in all stages in both years except flowering in 2020. Studies by Yadav et al. [34] indicated that foliar application of water could improve leaf hydration and nutrient absorption, stimulating photosynthesis and biomass production. However, ADM might not significantly increase with foliar spray of tap water compared to no spray due to the physiological shift towards reproductive processes during flowering in black gram unlike in 2021, which probably indicated a moisture deficit in 2020. Research by Sharma et al. [35] explored the dominance of reproductive demands over vegetative growth during this stage, limiting the impact of foliar watering on biomass accumulation. On the other hand, the significant increase in LAI with tap water spray compared to no spray during pod filling and maturity of black gram during 2021 can be attributed to improved leaf expansion, plant vigour and canopy development, potentially enhancing photosynthetic capacity and yield [36]. Co, K, and B nutrient spraying encouraged profuse branching, as well as the number and growth of leaves, resulting in increased biomass production and, consequently, the absorption of more PAR through an expanded photosynthesizing area [2].

Autumn-sown black gram's root nodulation started during its vegetative stage, whereas active nodulation emerged during the flowering stage. Active nodulation gradually decreases once the crop started their reproductive development. This was scientifically demonstrated by the current investigation, which showed a discernible decrease in the amount of black gram nodules during the pod growth stage. However, the individual application of each nutrient had a significant effect on the development of root nodules. This outcome was consistent with Sahay [37] and Singh [38] findings. Cobalt is a structural constituent of the co-enzyme cobalamin and methyl malonyl coenzyme A mutase [12, 33]. Hence, the element plays an important role in nitrogen fixation [39]. Additionally, the importance of both K and B are reported in numerous literatures [40, 41].

The leaf chlorophyll content of the plant is one of the fundamental attributing characters in connection with its photosynthetic capacity [42]. Soil-applied Co and foliar-applied K+B turned up to be extremely promising in accelerating the biosynthesis of chlorophyll while maintaining their structural integrity [43, 44]. This was quite evident from the higher values of total chlorophyll content associated with the corresponding treatments irrespective of the years. Despite substantial changes in ADM and LAI at 50% flowering stage with the intervention of any of the treatment in black gram, significant enhancements in net photosynthesis were found to be feasible in the current experiment. Studies like Sharma & Singh [45] have shown that improved chlorophyll content and stomatal conductance can boost photosynthetic rates, even without changes in biomass. Furthermore, Tiwari & Jha [46] demonstrated that efficient nutrient management can optimize resource allocation, fostering increased carbon assimilation. Thus, while traditional indicators may not reflect changes, physiological mechanisms elucidated in research highlighted the potential for notable improvements in net photosynthesis during critical growth stages of black gram. In the current experiment, the proper balance of nutrition for black gram with the active participation of applied plant nutrients seemed to help in optimum regulations of physiological and biochemical mechanisms [11]. In fact, earlier research efforts have elucidated the positive coordination of photosynthetic behaviour with improved plant mineral status or availability of nutrients to plants [47] together with greater chlorophyll biosynthesis [48]. Application of K [49] and B [50] also improved the photosynthetic pigment concentration.

Technically nitrate reductase is a responsible enzyme catalysing the reaction of nitrate reduction especially in the absence of biological nitrogen fixation with special reference to black gram [51]. In general, legumes noticeably reduce its rhizobial activity after flowering initiation [52]. This helps for the partitioning of the photo assimilates to the developing sink organs [10]. As the growth of the flower pods advances, the rate of the symbiotic nitrogen fixation process really tends to be almost zero. At that point, the plant is unable to convert atmospheric nitrogen into the necessary form of ammonium ($NH_4^+$) [53]. In the present experiment, the leaf sample for nitrate reductase analysis was obtained at the 50% flowering stage, when the black gram crop had to meet its internal nitrogen demand in nitrate form ($NO_2^+$) by extracting it from the soil reserve or from any external source. This requirement for nitrate nitrogen ($NO_2^+$) developed for its overall growth and development due to the indeterminate habit of growth, which could be fulfilled by subsequent N assimilation in the form of ammonium nitrogen ($NH_4^+$) through the process of nitrate reduction catalysed by nitrate reductase enzyme [54]. Notably, this assimilation is of great importance with respect to the quality enhancement of legume seeds in terms of protein content. In the present experiment, proper nutrient balance helped to optimize physiological and biochemical mechanisms thus maintaining considerable NR activity [11] at 50% flowering stage when the pod formation process was already initiated.

For net photosynthesis to occur at an efficient rate, solar radiation is crucial. A portion of incoming short-wave radiation falling within the 400–700 nm waveband is known as the PAR. Because of its absorption by pigments, primarily chlorophyll in plant leaves, and subsequent use in the process of carbon dioxide fixation in the photosynthesis system, this wave band, in particular, is photosynthetically active [2]. In the realm of agronomy, the consistent positive correlation between IPAR and LAI across all growth stages of black gram holds significant implications, accentuating the vital role of PAR in driving photosynthesis and subsequent leaf expansion, crucial for maximizing crop productivity [55]. This correlation reaffirms the importance of optimizing light interception efficiency for enhancing crop growth and yield in black gram cultivation [56]. In this context, the crop was treated with different levels of foliar spray at the very beginning of flowering stage as per the foliar treatment schedule, which could

not have significant impact regarding the %IPAR at flowering stage in both the years. However, the mentioned foliar nutrition significantly contributed to interception of PAR from pod filling stage onwards corresponding to the treatments, which was quite evident from Fig 4. Notably, PAR has been identified as a key factor influencing various physiological processes, particularly photosynthetic activity, which controls seed yield and total dry matter production [57]. In fact, IPAR and radiation use efficiency of the canopy for biomass production have been highlighted as the major determinants of different leguminous crops like mungbean [58], pigeon pea [59], grass pea [10] etc. As reported by a few researchers like Massignam et al. [60] and Sandana et al. [61], LAI as a major role in the intercept of solar radiation. Basu et al. [62] reported a similar increase in IPAR due to increasing LAI. Randawa [63] has also documented findings consistent with the same of the current experiment. These results clearly show the reason for the positive results in our studies

Improved PARUE is another important determining factor for an increased rate of photosynthesis [64]. It suggests that any crop can produce photosynthates for every unit of incoming solar radiation that strikes the crop canopy [33]. In this context, Worku and Demisie [59] recorded around 88% positive correlation between dry matter accumulation and radiation use efficiency regarding pigeon pea. Canopy structures, leaf area development, air temperature, water and nutrient availability can all influence the pattern and extent of radiation use efficiency [63]. This is reconcilable with the reports of Kumar et al. [65], Banerjee et al. [33] as well as Yahuza [66] in the case of soybean, blackgram and fababean respectively. A primary aspect thought to be responsible for any crop's increase in PARUE is optimal plant nutrition [67, 68].

Nutrient application whether by soil or foliar is found to significantly impact the growth and yield characteristics of any crop. We have also found the influence of nutrient management in improving growth and yield parameters of blackgram in the present experiment over the years. The influence of Co especially in legumes has been already reported earlier [69]. The influence of nutrients in enhancing the enzymatic activity positively triggering the nodulation is reported to be the reason for better germination and better yield [70]. Application of potassium and boron encouraged photosynthetic pigments production, flowering, nitrate reductase activity, pollen viability, flower to pod conversion and subsequent seed development to a greater extent which was quite evident from the higher yield [2]. These results were in closer agreement with those of Rao et al. [71]. Literatures have been found regarding Co, K and B related to their association with various enzymatic activities which evidently contributed to the development of qualitative and quantitative aspects of yield in legume crops [72–74]. Specifically, the unique nutrient management with Co, K and B might have attributed to better utilization of solar energy by means of facilitating profuse leaf production and canopy enlargement, improving its final biomass production along with LAI of black gram regardless of the experimental years [75]. This in turn, aided in acquiring greater photosynthetic area, improved photosynthesizing efficiency and better source-to-sink partitioning eventually leading to a spectacular increase in seed production [2, 33].

Plant nutrients are reported to influence the biochemical mechanisms at the cellular level improving the nutrient absorption capacity of the field crops [2, 39]. This may be the probable reason for the greater nutrient content in the seeds in the experiment we conducted. The protein content of the seed also gives us a clear indication of N accumulation in the seeds [76, 77]. The other beneficial role of these nutrients like Co for nutrient utilization [2, 78] K for water relation [79, 80] and B for effective biological N fixation [81] might have also influenced the balance between photosynthesis and transpiration.

Balanced nutrition is of utmost concern to attain higher production potential of black gram raised during the autumn season. Summarizing the results, it may be concluded that nutrient

management in terms of soil application of Co at 4 kg ha$^{-1}$ and exogenous foliar nutrition of 1.25% K coupled with 0.2% B at the flower initiation stage have immense potential to achieve substantial quality production in black gram by means of extensive canopy development, better symbiotic and physiological efficiency along with PAR interception to a great extent. This study details a special magnitude as a benchmark research for future researchers intending to upgrade the productivity level of autumn-sown rainfed black gram through judicious conjugation of beneficial with essential nutrients as a combined nutrient schedule.

## Institutional, national, and international guidelines and legislation

The materials and methods used in the study were performed in accordance with the relevant guidelines/regulations/legislation.

## Acknowledgments

The authors would like to sincerely thank to Bidhan Chandra Krishi Viswavidyalaya, Mohan-pur- 741252, Nadia, West Bengal, India; Bangladesh Wheat and Maize Research Institute, Dinajpur 5200, Bangladesh; ICAR-Central Research Institute for Dryland Agriculture, Hyder-abad 500059, Telangana, India and Taif University, Taif, Saudi Arabia to provide all facilities for successfully completed the study.

## Author Contributions

**Conceptualization:** Purabi Banerjee, Visha Kumari Venugopalan, Rajib Nath.

**Data curation:** Purabi Banerjee, Ahmed Gaber, Akbar Hossain.

**Formal analysis:** Purabi Banerjee, Ahmed Gaber, Akbar Hossain.

**Funding acquisition:** Rajib Nath, Ahmed Gaber.

**Investigation:** Purabi Banerjee, Visha Kumari Venugopalan, Rajib Nath.

**Methodology:** Purabi Banerjee, Visha Kumari Venugopalan, Rajib Nath.

**Resources:** Akbar Hossain.

**Software:** Purabi Banerjee, Ahmed Gaber, Akbar Hossain.

**Supervision:** Rajib Nath, Ahmed Gaber.

**Validation:** Purabi Banerjee, Visha Kumari Venugopalan, Rajib Nath.

**Visualization:** Purabi Banerjee, Visha Kumari Venugopalan, Rajib Nath.

**Writing – original draft:** Purabi Banerjee, Rajib Nath.

**Writing – review & editing:** Visha Kumari Venugopalan, Ahmed Gaber, Akbar Hossain.

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
