## [Decision Letter · Decision Letter 0]

24 Apr 2024

PONE-D-24-04425Dynamics of growth, physiology, radiation interception, production and quality of autumn black gram (Vigna mungo (L.) Hepper) as influenced by nutrient schedulingPLOS ONE

Dear Dr. Hossain,

Thank you for submitting your manuscript to PLOS ONE. After careful consideration, we feel that it has merit but does not fully meet PLOS ONE’s publication criteria as it currently stands. Therefore, we invite you to submit a revised version of the manuscript that addresses the points raised during the review process.

We look forward to receiving your revised manuscript.

Kind regards,

Banwari Lal, Ph.D.

Academic Editor

PLOS ONE

Journal Requirements:

The researchers would also like to acknowledge the Bangladesh Wheat and Maize Research Institute, Dinajpur, Bangladesh and Deanship of Scientific Research, Taif University for funding this work.

The researchers would also like to acknowledge the Deanship of Scientific Research, Taif University for funding this work.

The researchers would also like to acknowledge the Bangladesh Wheat and Maize Research Institute, Dinajpur, Bangladesh and Deanship of Scientific Research, Taif University for funding this work.

Additional Editor Comments:

Section/Line No Comments

Introduction: Can you provide more context on why cobalt, potassium, and boron were selected as the focus nutrients for this study? Were there specific deficiencies or growth limitations observed in previous research that prompted their investigation?

The role/importance of studied parameter “nitrate reductase” is missing from introduction. Mention it

Material and methodology Why photosynthetic and nitrate reductase was studied at 50% flowering stage only? ADM and LAI are not at all affected at this stage by any treatment but still you are reporting significant values in table 3? explain

Figure 2: In Figure 2 and all subsequent figures, is there any specific reason for maintaining large gap between “cobalt” and “no spray” treatments. Keep uniform gap among the treatments in all figures.

Figure 2A and B: Why ADM in 2020 is significantly higher in “tap water” compared to “No spray” in all stages except flowering where it is comparable? Moreover, in next year (2021) it remains comparable in almost all stages. explain and justify

Yearwise, bar colours are different for the same stage (For example, Maturity stage in 2020 denoted by red colour but in 2021 it is red. Use uniform pattern in all figures.

Mention other results of other treatments (K, B) as well in text.

Figure 3B: Why LAI in 2021 is significantly higher in “tap water” compared to “No spray” in pod filling and maturity?

L61-62: Statement is “The interception of solar radiation holds an immense significance on developmental aspects of crop plants including food legumes [1,2]” However quoted ref [1,2] are about “WHEAT” a cereal and NOT legume as misquoted by author

L140: In experimental design and treatment details, why the abbreviations for each treatment are mentioned when they are not used in text and figure for even single time? Remove it

L172: Mention the formula equation for calculating aerial dry matter.

L187-88: Write procedure in details and name of the substate used for nitrate reductase activity

Table 2: Make table year wise, as figures are made. In present format it is difficult to compare between stages.

Why data is not analysed stage-wise as in figures.

L244-48: Rewrite after analysing stagewise data.

Figure 4: Explain why there in no significant effect of treatments Flowering stage of both year but immediately in next stage i.e. pod filling there is exactly same pattern in 2020 and 2021

Figure 5: In caption it is mentioned that “Error bars indicate the standard error of the mean. Different alphabets delineate significant differences between means” but in figure there is no error bars and alphabets are present. Correct it

Figure 6: Describe results in text with more details

L221-25: This is irrespective to the treatments. No need to mention it.

L225-27: Explain year wise data for better understanding.

L228-29: Statement “ …K+B recorded maximum ADM and LAI…”. Its false statement, review it (as ADM and LAI are statistically same in cobalt and K+B treatments.

L229: Statement “……which were statistically significant as well”. Mention needed for compared to which treatment (s)?

L271: Remove extra full stop.

L288-89: State whether it is significant or non-significant and justify accordingly

L403: Statement “Randawa [48] has also reported similar results as we have reported” rewrite

L403: In text ref “Randawa” has given [48] number in literature cited at 48 number same reference is not mentioned.

Reference Please see for uniform formatting as per journal style

Reviewers' comments:

Reviewer's Responses to Questions

**Comments to the Author**

1. Is the manuscript technically sound, and do the data support the conclusions?

Reviewer #1: Partly

2. Has the statistical analysis been performed appropriately and rigorously? 

Reviewer #1: Yes

3. Have the authors made all data underlying the findings in their manuscript fully available?

Reviewer #1: No

4. Is the manuscript presented in an intelligible fashion and written in standard English?

Reviewer #1: Yes

5. Review Comments to the Author

**Reviewer #1:** The “Dynamics of growth, physiology, radiation interception, production and quality of

autumn black gram (Vigna mungo (L.) Hepper) as influenced by nutrient scheduling” is a good study and good information is being generated and well presented. There is need to address following comments before publication.

Section/Line No Comments

Introduction: Can you provide more context on why cobalt, potassium, and boron were selected as the focus nutrients for this study? Were there specific deficiencies or growth limitations observed in previous research that prompted their investigation?

The role/importance of studied parameter “nitrate reductase” is missing from introduction. Mention it

Material and methodology Why photosynthetic and nitrate reductase was studied at 50% flowering stage only? ADM and LAI are not at all affected at this stage by any treatment but still you are reporting significant values in table 3? explain

Figure 2: In Figure 2 and all subsequent figures, is there any specific reason for maintaining large gap between “cobalt” and “no spray” treatments. Keep uniform gap among the treatments in all figures.

Figure 2A and B: Why ADM in 2020 is significantly higher in “tap water” compared to “No spray” in all stages except flowering where it is comparable? Moreover, in next year (2021) it remains comparable in almost all stages. explain and justify

Yearwise, bar colours are different for the same stage (For example, Maturity stage in 2020 denoted by red colour but in 2021 it is red. Use uniform pattern in all figures.

Mention other results of other treatments (K, B) as well in text.

Figure 3B: Why LAI in 2021 is significantly higher in “tap water” compared to “No spray” in pod filling and maturity?

L61-62: Statement is “The interception of solar radiation holds an immense significance on developmental aspects of crop plants including food legumes [1,2]” However quoted ref [1,2] are about “WHEAT” a cereal and NOT legume as misquoted by author

L140: In experimental design and treatment details, why the abbreviations for each treatment are mentioned when they are not used in text and figure for even single time? Remove it

L172: Mention the formula equation for calculating aerial dry matter.

L187-88: Write procedure in details and name of the substate used for nitrate reductase activity

Table 2: Make table year wise, as figures are made. In present format it is difficult to compare between stages.

Why data is not analysed stage-wise as in figures.

L244-48: Rewrite after analysing stagewise data.

Figure 4: Explain why there in no significant effect of treatments Flowering stage of both year but immediately in next stage i.e. pod filling there is exactly same pattern in 2020 and 2021

Figure 5: In caption it is mentioned that “Error bars indicate the standard error of the mean. Different alphabets delineate significant differences between means” but in figure there is no error bars and alphabets are present. Correct it

Figure 6: Describe results in text with more details

L221-25: This is irrespective to the treatments. No need to mention it.

L225-27: Explain year wise data for better understanding.

L228-29: Statement “ …K+B recorded maximum ADM and LAI…”. Its false statement, review it (as ADM and LAI are statistically same in cobalt and K+B treatments.

L229: Statement “……which were statistically significant as well”. Mention needed for compared to which treatment (s)?

L271: Remove extra full stop.

L288-89: State whether it is significant or non-significant and justify accordingly

L403: Statement “Randawa [48] has also reported similar results as we have reported” rewrite

L403: In text ref “Randawa” has given [48] number in literature cited at 48 number same reference is not mentioned.

Reference Please see for uniform formatting as per journal style

6. PLOS authors have the option to publish the peer review history of their article (what does this mean?). If published, this will include your full peer review and any attached files.

Reviewer #1: **Yes: **Dr Shankar Lal Jat

---

## [Author Response · Author response to Decision Letter 0]

8 May 2024

Date: 08 May 2024

To

Banwari Lal, Ph.D.

Academic Editor: PLOS ONE

Ref: PONE-D-24-04425

Title: Dynamics of growth, physiology, radiation interception, production and quality of autumn black gram (Vigna mungo (L.) Hepper)) as influenced by nutrient scheduling

Sub: Submission of the revised version of the article ‘PONE-D-24-04425’ after resolving all issues raised by the editor and reviewers’

Dear Editor,

Thank you for sending the reviewers’ and editorial comments, which have allowed us to further improve the article. We are happy to inform you that we have been able to resolve all aspects raised by the editor and reviewers. All edits are shown in yellow, as well as the track change mode in the text of the article. Unfortunately, we have overlooked any aspects that kindly let us know, and we will rectify that immediately. Please see our response to all the comments below:

Response to Editorial and reviewers’ comments

Journal Requirements

Comment 1. When submitting your revision, we need you to address these additional requirements.

Authors’ Response: The current version of the article is rearranged as per style of the journal

Comment 2. Thank you for stating the following financial disclosure:

The researchers would also like to acknowledge the Bangladesh Wheat and Maize Research Institute, Dinajpur, Bangladesh and Deanship of Scientific Research, Taif University for funding this work.

Authors’ Response: The financial disclosure section should be revised as: The study was financially supported by Bidhan Chandra Krishi Viswavidyalaya, Mohanpur- 741252, Nadia, West Bengal, India; Bangladesh Wheat and Maize Research Institute, Dinajpur 5200, Bangladesh and ICAR-Central Research Institute for Dryland Agriculture, Hyderabad 500059, Telangana, India. The research was also funded by Taif University, Saudi Arabia, Project No. (TU-DSPP-2024-07).

We are also requesting you to incorporate a statement in the section. For example, ‘The funders had no role in study design, data collection and analysis, decision to publish, or preparation of the manuscript’.

Comment 3: Thank you for stating the following in the Acknowledgements Section of your manuscript: The researchers would also like to acknowledge the Deanship of Scientific Research, Taif University for funding this work.

We note that you have provided funding information that is not currently declared in your Funding Statement. However, funding information should not appear in the Acknowledgements section or other areas of your manuscript. We will only publish funding information present in the Funding Statement section of the online submission form.

The researchers would also like to acknowledge the Bangladesh Wheat and Maize Research Institute, Dinajpur, Bangladesh and Deanship of Scientific Research, Taif University for funding this work.

Authors’ Response: The acknowledgement section should be modified as: The authors would like to sincerely thank to Bidhan Chandra Krishi Viswavidyalaya, Mohanpur- 741252, Nadia, West Bengal, India; Bangladesh Wheat and Maize Research Institute, Dinajpur 5200, Bangladesh and ICAR-Central Research Institute for Dryland Agriculture, Hyderabad 500059, Telangana, India to provide all facilities for successfully completed the study. Authors would also appreciate the support and facility of Taif University, Taif, Saudi Arabia under project number (TU-DSPP-2024-07).

Comments of editorial/Reviewer 1 

Comment: The “Dynamics of growth, physiology, radiation interception, production and quality of

autumn black gram (Vigna mungo (L.) Hepper) as influenced by nutrient scheduling” is a good study and good information is being generated and well presented. There is need to address following comments before publication.

Authors’ Response: Authors are pleased to note the constructive suggestions and positive remarks from Reviewer 1. All the criteria of the reviewer have been minutely taken into consideration and changes have been incorporated in the revised manuscript accordingly wherever applicable.

Comment: Introduction: Can you provide more context on why cobalt, potassium, and boron were selected as the focus nutrients for this study? Were there specific deficiencies or growth limitations observed in previous research that prompted their investigation?

Authors’ Response: The context has been elaborated in line no 101-109 in the revised manuscript.

Comment: The role/importance of studied parameter “nitrate reductase” is missing from introduction. Mention it.

Authors’ Response: The role/importance of studied parameter “nitrate reductase” has been included in line no 92-96 under introduction in the revised manuscript.

Comment: Material and methodology: Why photosynthetic and nitrate reductase was studied at 50% flowering stage only?

Authors’ Response: The explanations have been included in line no 201-207 and 222-227, respectively in the revised manuscript.

Comment: ADM and LAI are not at all affected at this stage by any treatment but still you are reporting significant values in table 3? explain

Authors’ Response: The fact has been discussed with proper explanation matter in line no 460-468 in the revised manuscript.

Comment: Figure 2: In Figure 2 and all subsequent figures, is there any specific reason for maintaining large gap between “cobalt” and “no spray” treatments. Keep uniform gap among the treatments in all figures.

Authors’ Response: A specific gap has been maintained between “cobalt” and “no spray” treatments in every case which is a little longer than the gaps between any two levels of the individual factors just to create a visual ease for differentiating the two separate factors of the experiment with corresponding levels, while keeping uniform gap between the different levels of any of the factors.

Comment: Figure 2A and B: Why ADM in 2020 is significantly higher in “tap water” compared to “No spray” in all stages except flowering where it is comparable? Moreover, in next year (2021) it remains comparable in almost all stages. explain and justify.

Authors’ Response: The explanation and justification have been added matter in line no 433-440 in the revised manuscript.

Comment: Yearwise, bar colours are different for the same stage (For example, Maturity stage in 2020 denoted by red colour but in 2021 it is red. Use uniform pattern in all figures.

Authors’ Response: Uniform bar colours have been maintained for same stage in both year in Figure 2 and 3.

Comment: Mention other results of other treatments (K, B) as well in text.

Authors’ Response: The outcomes for foliar K spray and B spray have been included in the results wherever applicable.

Comment: Figure 3B: Why LAI in 2021 is significantly higher in “tap water” compared to “No spray” in pod filling and maturity?

Authors’ Response: The explanation and justification have been added matter in line no 440-443 in the revised manuscript.

Comment: L61-62: Statement is “The interception of solar radiation holds an immense significance on developmental aspects of crop plants including food legumes [1,2]” However, quoted ref [1,2] are about “WHEAT” a cereal and NOT legume as misquoted by author

Authors’ Response: Citation has been modified which are exclusively into pulse crops.

Comment: L140: In experimental design and treatment details, why the abbreviations for each treatment are mentioned when they are not used in text and figure for even single time? Remove it

Authors’ Response: Treatment abbreviations have been removed.

Comment: L172: Mention the formula equation for calculating aerial dry matter.

Authors’ Response: Formula equation has been added for calculating aerial dry matter in line no 187 in the revised manuscript.

Comment: L187-88: Write procedure in details and name of the substate used for nitrate reductase activity

Authors’ Response: Detailed procedure of NR estimation with substrate used have been included in line no 210-221 in the revised manuscript.

Comment: Table 2: Make table year wise, as figures are made. In present format it is difficult to compare between stages.

Authors’ Response: All the three tables have been modified into year wise format.

Comment: Why data is not analysed stagewise as in figures. L244-48: Rewrite after analysing stagewise data.

Authors’ Response: The portion has been rewritten after analysing stagewise data in line no 301-307 in the revised manuscript.

Comment: Figure 4: Explain why there in no significant effect of treatments Flowering stage of both year but immediately in next stage, i.e., pod filling there is exactly same pattern in 2020 and 2021

Authors’ Response: The explanation has been added matter in line no 499-503 in the revised manuscript.

Comment: Figure 5: In caption it is mentioned that “Error bars indicate the standard error of the mean. Different alphabets delineate significant differences between means” but in figure there is no error bars and alphabets are present. Correct it

Authors’ Response: The sentence has been deleted from the caption of Figure 5.

Comment: Figure 6: Describe results in text with more details

Authors’ Response: Detailed results of Figure 6 have been presented in line no 379-389 in the revised manuscript.

Comment: L221-25: This is irrespective to the treatments. No need to mention it.

Authors’ Response: The portion has been removed.

Comment: L225-27: Explain year wise data for better understanding.

Authors’ Response: The segment has been modified in line no 264-278 in the revised manuscript.

Comment: L228-29: Statement “ …K+B recorded maximum ADM and LAI…”. Its false statement, review it (as ADM and LAI are statistically same in cobalt and K+B treatments.

Authors’ Response: Basically, soil application and foliar spray are two separate factors in this experiment. The analysis has been presented separately for these two factors in each table, under which, the respective levels are under comparison. Therefore, there is no comparison as such between soil cobalt application and K+B foliar spray. For more clarification, the sentence mentioned has been rephrased in line no 273-278 in the revised manuscript.

Comment: L229: Statement “……which were statistically significant as well”. Mention needed for compared to which treatment (s)?

Authors’ Response: Treatment comparison has been mentioned in line no 273-278 in the revised manuscript.

Comment: L271: Remove extra full stop.

Authors’ Response: Extra full stop has been removed.

Comment: L288-89: State whether it is significant or nonsignificant and justify accordingly.

Authors’ Response: The significance has been justified in line no 495-499 in the revised manuscript.

Comment: L403: Statement “Randawa [48] has also reported similar results as we have reported” rewrite

Authors’ Response: The sentence has been rewritten in line no 510-511 in the revised manuscript.

Comment: L403: In text ref “Randawa” has given [48] number in literature cited at 48 number same reference is not mentioned.

Authors’ Response: Citation number has been modified.

Comment: Reference, please see for uniform formatting as per journal style

Authors’ Response: Entire reference section has been checked and formatted.

In addition to these corrections, we have carefully rechecked the entire article and modified it per the journal’s requirements. Please check the main text of the revised manuscript for responsewise modifications.

We look forward to your editorial decision.

With regards

Akbar Hossain (Corresponding author)

http://orcid.org/0000-0003-0264-2712

---

## [Editor Report · Decision Letter 1]

14 May 2024

Dynamics ofgrowth, physiology, radiation interception,production and quality of autumn black gram ( Vigna mungo (L.) Hepper) as influenced by nutrient scheduling

PONE-D-24-04425R1

Dear Dr. Hossain,

We’re pleased to inform you that your manuscript has been judged scientifically suitable for publication and will be formally accepted for publication once it meets all outstanding technical requirements.

Kind regards,

Banwari Lal, Ph.D.

Academic Editor

PLOS ONE
---

## [Editor Report · Acceptance letter]

22 Aug 2024

PONE-D-24-04425R1 

PLOS ONE

Dear Dr. Hossain, 

I'm pleased to inform you that your manuscript has been deemed suitable for publication in PLOS ONE. Congratulations! Your manuscript is now being handed over to our production team.

Kind regards, 

on behalf of

Dr. Banwari Lal 

Academic Editor

PLOS ONE